# DEP-on-a-Chip: Dielectrophoresis Applied to Microfluidic Platforms

**DOI:** 10.3390/mi10060423

**Published:** 2019-06-24

**Authors:** Haoqing Zhang, Honglong Chang, Pavel Neuzil

**Affiliations:** 1Ministry of Education Key Laboratory of Micro/Nano Systems for Aerospace, School of Mechanical Engineering, Northwestern Polytechnical University, 127 West Youyi Road, Xi’an 710072, China; zhanghaoqing@mail.nwpu.edu.cn; 2Central European Institute of Technology, Brno University of Technology, Brno 61300, Czech Republic; 3Department of Microelectronics, Faculty of Electrical Engineering, Brno University of Technology, Technická 3058/10, Brno 61600, Czech Republic

**Keywords:** dielectrophoresis, microfluidics, two-dimensional electrodes, parallel electrodes, interdigitated electrode, castellated electrodes, three-dimensional electrodes

## Abstract

Dielectric particles in a non-uniform electric field are subject to a force caused by a phenomenon called dielectrophoresis (DEP). DEP is a commonly used technique in microfluidics for particle or cell separation. In comparison with other separation methods, DEP has the unique advantage of being label-free, fast, and accurate. It has been widely applied in microfluidics for bio-molecular diagnostics and medical and polymer research. This review introduces the basic theory of DEP, its advantages compared with other separation methods, and its applications in recent years, in particular, focusing on the different electrode types integrated into microfluidic chips, fabrication techniques, and operation principles.

## 1. Introduction

The establishment of lab-on-a-chip (LOC) technology made it possible to gradually integrate large and costly bench-top laboratory instruments into palm-sized devices containing chips in a scale of a few centimeters or less, without sacrificing their performance [1]. LOC-based devices have been used in molecular biology [2], clinical diagnostics [3], and point-of-care systems [4].

The fabrication of microfluidic chips requires a demanding technological process to integrate a variety of structures and components to perform different processes, such as sample pretreatment [5], sample delivery [6], particle manipulation [7], on-chip reaction [8], and result detection and analysis [9]. Particle separation [10], a subcategory of particle manipulation, is a crucial process affecting detection precision. Many techniques have so far been used for this, such as filtration [11,12], centrifugation [13], the use of magnetic [12,14] and acoustic force [15,16], chromatography [17,18], electrophoresis [19,20], and dielectrophoresis (DEP) [21,22]. Compared to other methods, DEP is becoming one of the most promising separation techniques for micro- and nano-scale systems because of its low running cost, ease of integration into microfluidics, speed, efficiency, sensitivity, and selectivity. Finally, it is also a label-free method [23], making sample processing very simple.

Here, we present the theory of DEP, a comparison between DEP and other separation methods, and an explanation of its applications in microfluidic chips. We review the advantages and disadvantages of major systems, their impact, and potential, in particular, focusing on different electrode types integrated into microfluidic chips with structures, fabrication techniques, and operation principles. The brief review will be of interest to researchers in DEP microfluidics.

## 2. Microfluidic Separation Techniques

Particle separation is a technique that extracts or obtains target particles from a solution or suspension or isolates them from each other based on differences in the particles’ physical and or chemical properties or differences between the properties of the particles and the carrier fluid. It can also be considered as a concentration or purification process for target chemicals or particles.

Filtration [24,25,26] is a commonly used and effective separation method based on particle size differences. A porous medium with a specific pore size forms a filter, allowing particles smaller than the pore size, as well as the carrier fluid, to pass through the filter, where oversized particles are collected. However, it is challenging to apply filtration to microfluidic chips once the porous medium becomes blocked during the filtration process, thus gradually decreasing the throughput unless a more complex ultrafiltration system is used [27]. These disadvantages or complexities hinder the wide employment of this technique for LOC systems.

The centrifugation method [28,29,30,31,32] is used to separate particles based on specific mass while being subjected to centrifugal force. However, it requires the application of an additional tool to generate the force.

Magnetic force separation [33,34,35,36] mainly depends on the magnetic susceptibility of the particles. It is possible to bind magnetic beads with the particles of interest, but this increases the system’s complexity.

Acoustic tweezers [37,38,39,40] are a separation method based on sound waves. Particles can be induced to move toward acoustic pressure nodes or pressure antinodes in a standing acoustic field by an acoustic-radiation force. The movements are dependent on the particles’ density, and precise movement can be obtained by controlling the position of these nodes. However, the precision and throughput limit the application of acoustic tweezers for LOC systems.

Chromatography [41,42,43,44] is a powerful separation technique widely applied in microfluidic platforms, where the mobile phase carries the target particles and passes through a stationary phase. Different particles in the mixture travel at different speeds, resulting in their separation. However, the chromatography in microfluidic chips require complex structures, including valves, pumps, and stationary phases, subsequently increasing its fabrication complexity. The requirement of sample labeling and especially sample loss due to unspecific adhesion to the large surface area of the stationary phase prevents its wide application.

Electrophoresis [45,46,47] is a technique to separate charged particles in the suspension based on a uniform electric field and is widely applied in the study of deoxyribonucleic acids (DNA), ribonucleic acids, and proteins. However, it is only available for charged particles, limiting its potential application.

In comparison with other separation methods, DEP [48,49,50] has the unique advantage of being label-free, fast, and accurate, which is why it has been widely applied in microfluidics for bio-molecular diagnostics, and medical and polymer research.

## 3. Dielectrophoresis

DEP is a method used to separate suspended particles from the suspension or carrier fluid by generating polarization forces in a non-uniform electric field (NUEF) [51]. Converse to electrophoresis, the particles in DEP do not carry electrical charges, but they must be polarizable. These particles that are exposed to an electric field and become polarized forming electric dipoles; the resulting force on these dipoles is called DEP force (*F*_DEP_). *F*_DEP_ is 0 N (Figure 1A) when the particle is in a uniform electric field, while in NUEF, the force is either positive (pDEP) or negative (nDEP) making the particle move in a certain direction depending on its polarization (Figure 1B) [51].

The amplitude of *F*_DEP_ of a spherical particle in the suspension follows Equations (1) and (2) [52].(1)$$F_{\mathrm{DEP}}=2\pi \varepsilon_{m}\varepsilon_{0}r_{\mathrm{ext}}^{3}Re\left[CM\left(f\right)\right]\nabla E_{\mathrm{RMS}}^{2}$$ where *ε*_m_ is the medium relative permittivity, *ε*_0_ is the vacuum permittivity, *r*_ext_ is a radius of the spherical particle, *E*_RMS_ is the root-mean-square value of the applied electric field, and *CM*(*f*) is the Clausius–Mossotti factor.(2)$$CM\left(f\right)=\frac{\varepsilon_{p}^{*}-\varepsilon_{m}^{*}}{\varepsilon_{p}^{*}+2\varepsilon_{m}^{*}}$$ where $\varepsilon_{p}^{*}$ and $\varepsilon_{m}^{*}$ are defined as:(3)$$\varepsilon_{p}^{*}=\varepsilon_{p}\varepsilon_{0}-j\frac{\sigma_{p}}{\pi f},\;\varepsilon_{m}^{*}=\varepsilon_{m}\varepsilon_{0}-j\frac{\sigma_{m}}{\pi f}$$ where *ε*_p_ is the relative permittivity of a particle, *σ*_p_ is the electric conductivity of a particle, *σ*_m_ is the electric conductivity of the medium, and *f* is the frequency of the electric field.

The NUEF is powered by the direct current (DC) or alternating current (AC) and is generated by geometry of insulating posts in combination with channel tapering, etc., in the microfluidic channel [10] also known as AC/DC-iDEP [53]. The microfluidic chips based on DC-iDEP have a simple structure with no gas bubble generation during particle separation. However, the power loss in the DC electrical field results in Joule heat generation, excessively increasing the temperature inside the device. Compared to this, the AC-DEP is more widely used because the Joule heat generation is greatly suppressed. The device selectivity then depends on the AC frequency as well as the field strength; thus, the parameters of both should be optimized.

## 4. Dielectrophoresis (DEP)-on-a-Chip

A variety of DEP microfluidic devices have been developed for use in bioparticle separation. The gradient of the electric field is an important factor that affects the DEP performance. The electrodes’ geometry mainly produces the NUEF in the microfluidic chip, so the shape and distribution of the electrodes are vital to the NUEF amplitude. Here we discuss different electrodes applied in microfluidics approaches, such as DEP performed by external electrodes and 2D and 3D electrodes.

### 4.1. External Electrodes

Researchers have developed an iDEP microfluidic chip made of polydimethylsiloxane (PDMS) covered with a glass lid. The device’s performance was evaluated in a hydrodynamic study of blood alteration [54]. The chip structure consisted of an inlet and an outlet port, and a ≈ 1 cm long, ≈ 50 μm deep main channel with 38 dead-end branches distributed evenly on both sides of the main channel (Figure 2A). The width and depth of the branches were ≈ 200 μm and ≈ 50 μm, respectively. An electric field was introduced into the device by inserting two Pt electrodes, one into the inlet and one into the outlet. A sample of blood with a volume of ≈ 2 μL was injected into the main channel and pumped through it using internal capillary forces. The branches hydro-dynamically trapped the red blood cells (RBCs). Then, the DC power supply was activated to establish the DC-DEP force, preventing more RBCs from being trapped inside the dead-end channels near the inlet port. Additionally, the electro-osmotic flow generated by the DC electric field removed RBCs from the cross junctions formed at the channels and branches. As a result, the plasma was separated from the blood, forming a plasma area without RBCs. This method obtained a plasma purity of 99%. The iDEP device performed the plasma separation without diluting the blood samples. This device was used with lower voltage in comparison to other iDEP devices [55].

A simple PDMS microfluidic chip was fabricated using soft lithography to perform the wall-induced DEP [56] (Figure 2B). This device was used to separate oil droplets from Janus particles, having two or more distinct physical properties on the surface covered with oil phase. The suspension containing the particles of interest was moving from the sample inlet to the main channel, and the buffer from the sheath fluid inlet pushed the particles to move along the wall of the main channel. The length and width of the sample inlet and sheath fluid inlet were ≈ 5 mm and ≈ 150 μm, ≈ 1 mm, and ≈ 250 μm, respectively. The sample inlet was next to a confinement chamber of ≈ 6 mm × 6 mm. The length of both outlets was ≈ 9 mm, and their widths were ≈ 120 μm and ≈ 180 μm, respectively. The depth of the confinement chamber was ≈ 80 μm, allowing only droplets of a diameter of < 80 μm to pass through the channel. The main channel, with a length and depth of ≈ 1 mm and ≈ 250 μm, respectively, connected the inlets and outlets. The channel tapering generated the NUEF after four Pt electrodes were inserted into the inlets and outlets. The deionized water first wetted the channel, and then the Janus droplets covered by oil droplets were injected into the confinement chamber and pushed close to the main channel by the fluid from the sheath fluid inlet. The DC power supply was then applied to the channel, and the Janus and oil droplets were gradually separated from each other with the help of the *F*_DEP_ and moved into the different outlets. The separation process was dependent on the amplitude of the applied voltage and droplet size. The fabrication process was rather simple because the electrodes were inserted into the chip and not monolithically integrated into the device.

### 4.2. Two-Dimensional Electrodes

Thin-film 2D electrodes based on a simple fabrication process with different shapes were developed to meet the requirements of integration and to increase DEP separation efficiency within the microfluidic chips. Here, we introduce the most commonly used electrode configurations, parallel [52,57,58,59,60] and castellated electrodes [61,62,63,64], and briefly present other such configurations, such as quadrupole electrodes [65,66].

#### 4.2.1. Parallel and Interdigitated Electrodes

The electrodes from both parallel and interdigitated electrode arrays are typically rectangular. The electrodes have identical widths and gaps resulting in a constant electric field between them. Thus, the system based on them has a high separation efficiency of particles of interest.

A microfluidic device was proposed to separate green fluorescent protein labeled MDA-MB-231, and other types of cancer cells from healthy ones [52]. The microfluidic chip consisted of two parts (Figure 3A). First, the electrode part was made using a glass wafer as a substrate, which was sputtered and coated with a sandwich layer of Cr/Au. This sandwich metal layer was then patterned using a wet etching process to form a planar interdigitated electrode array. This consisted of 20 pairs of electrodes with widths and gaps of ≈ 39 μm and ≈ 36 μm, respectively, protruding into the channel by ≈ 2 µm to ≈ 3 µm, providing powerful NUEF. Next, the microfluidic structure was formed using PDMS and a soft lithography technique. The width of the channel was ≈ 50 μm and it was enlarged to ≈ 250 μm. Finally, these two parts, the electrodes and microfluidic structure, were bonded together, assisted by an oxygen plasma. The sample was pumped into the channel, and the NUEF was formed by the AC power supply with a set amplitude of 15 V peak to peak (V_PP_) at a set frequency of 40 kHz, obtaining a separation accuracy and purity of ≈ 100% and ≈ 81%, respectively.

A microfluidic chip was fabricated to combine particles and DEP to perform continuous separation of yeast cells from silica particles [60] by ICEO, using a microscope glass slide as the device substrate. The glass was coated with a thin layer of indium tin oxide (ITO) and subsequently patterned, forming parallel electrodes. The channels, one inlet and four outlets, were made in the PDMS (Figure 3B) and bonded to the glass substrate using an oxygen plasma surface-activation method. The microfluidic chip had three regions: focusing, transition, and separation. First, the suspension, consisting of silica particles, yeast cells, and carrier fluid, was injected into the chip, and the ICEO focused the particles in the focusing region. The side branches, with optimized size in the transition region, ensured the particle stream flowed into the separation region in a straight line. The separation efficiency was > 96% with the AC power supply at a set value of 225 V_PP_ at 1 MHz. This general-purpose DEP microfluidic chip has great potential in molecular diagnostics in LOC systems because it exhibited high throughput. The integration of the ICEO module simplified the chip structure, as there is no need for a complicated flow rate control module and narrow microfluidic channel to focus the particle. It provided an alternative method to focus the particles, resulting in an integrated device.

A microfluidic platform with a hybrid DEP-inertial system was developed to separate particles [64] (Figure 3C). The channel, which was ≈ 200 μm wide and ≈ 40 μm deep, consisted of 15 symmetric U-shaped segments with a length of ≈ 700 μm and was patterned in PDMS using soft lithography. The inlet and outlet holes were also punched in the PDMS. A lift-off technique patterned the thin-film Ti (50 nm)/Pt (150 nm) electrodes on a glass slide in a zigzag distribution. The gap and width of the electrodes were both ≈ 20 μm. The bonding between PDMS and the glass slide was performed by oxygen plasma. A suspension flow of polystyrene beads with diameters of ≈ 5 μm and ≈ 13 μm was injected from the inlet, and the sheath flow from the top pushed the particles along the bottom of the channel, keeping them in the NUEF. The field strength of the AC power supply was optimized to ≈ 27 V_PP_ at ≈ 1 MHz, obtaining a separation efficiency of ≈ 13 μm and ≈ 5 μm particles of ≈ 100% and ≈ 96%, respectively. The varied particle size separations were achieved inside this chip by adjusting the voltage without needing a redesign of the chip layout. The implementation of the top sheath flow improved the separation efficiency as the particles were forced to the bottom of the channel and were not influenced by the electric field decay along the vertical direction.

An isomotive DEP microfluidic device was developed to separate living and dead yeast cells (Figure 4) [67]. The device consisted of three sections: three inlets, a separation section, and two outlets. The electrodes for DEP were made of a metal sandwich of Ti/Pt/Au with a thickness of ≈ 50 nm, ≈ 50 nm, and ≈ 300 nm, respectively, on a glass substrate. The electrode width and gap between them were both ≈ 10 μm. The fluidic channels were patterned on top of the electrodes using SU-8 photoresist with a thickness of ≈ 100 μm and then sealed by a glass coverslip. The length and width of the main channel were ≈ 600 μm and ≈ 100 μm, respectively. The side of the coverslip facing the chamber was coated with ITO, serving as a ground electrode. The electrode biasing network was formed using the resistor array next to the main channel and was connected to the electrodes to perform the isomotive DEP.

The suspension containing yeast cells was pumped into the main channel from the middle inlet, and the cells in suspension were focused to move in a streamline with the help of sheath flow generated by the buffer from the other two side channels. The living and dead yeast cells were separated in the DEP section under a non-uniform isomotive electric field. The device performance was evaluated by applying AC power supply with a set value of 14 V_PP_ at different frequencies.

A microfluidic device with interdigitated electrodes was fabricated to separate living and dead *Listeria* cells (Figure 5A) [68]. The Cr/Pt with a thickness of ≈ 10 nm and 100 nm, respectively, were sputtered using a magnetron system on a glass substrate, and then the interdigitated electrodes were patterned using the photolithography method with subsequent etching. The gap and the width of the electrodes were both ≈ 15 μm. Then, a rectangular chamber was constructed of silicon rubber with a thickness of ≈ 0.3 mm, and a ≈ 20 μL suspension containing the *Listeria* cells was pipetted into the chamber. A glass coverslip was then used to seal the chamber after loading the suspension. The AC power supply was applied with a set value of 1 V_PP_ from 10 kHz to 15 MHz, and the cell separation performance was monitored through a charge-coupled device camera. The separation efficiency of living and dead *Listeria* cells was > 90% at a set frequency of 50 kHz. This system combined antibody recognition and DEP and thus improved the selectivity of capturing bacteria.

A DEP microfluidic device with interdigitated electrodes was developed to concentrate and selectively capture *Listeria* cells bonding with antibodies (Figure 5B) from other types of cells [69]. The interdigitated electrodes were made of a Ti/Pt sandwich with a thickness of ≈ 20 nm and ≈ 80 nm, respectively, by a sputtering and a lift-off process on an oxidized silicon substrate. The width and gap of the electrodes were ≈ 23 μm and ≈ 17 μm, respectively. The electrodes were then covered with a SiO_2_ layer with a thickness of ≈ 300 nm using plasma-enhanced chemical-vapor deposition, preventing electro-osmotic currents near the electrodes via the formation of a bimetallic electrochemical cell. The PDMS patterned with fluidic structure was bonded to the substrate using an oxygen plasma enhancement process at ≈ 200 W for ≈ 10 s. The length, width, and depth of the channel were ≈ 3000 μm, ≈ 260 μm, and ≈ 16 μm, respectively. The tubes were inserted into the inlet and outlet to deliver the suspension into the channel. The AC power supply was applied with a set value of 20 V_PP_ at 1 MHz to capture the *Listeria* cells bonding with antibodies from the suspension. The result showed that > 90% *Listeria* cells bonding with antibodies were collected with a flow rate of ≈ 0.2 μL·min^−^^1^, and the increased flow rate decreased the capture efficiency.

#### 4.2.2. Castellated Electrodes

The castellated electrodes generated an electrical field gradient, forming a maximum value at the edges of the microelectrodes and a minimum value at the center between two electrodes.

A microfluidic device that combined an insulator with 2D electrodes was developed to focus and separate particles (Figure 6A,B) [70]. The electrodes were designed in a parallel configuration with a total number of 30 electrodes having 15 electrodes on each side of the main channel. The metal sandwich of Ti/Pt was patterned using a lift-off technique on a Pyrex glass substrate. The fluidic structure was made in a SU-8 layer with a thickness of ≈ 20 µm, forming a castellated structure (Figure 6A). A PDMS cover with access holes was mounted on top of the structure to seal it. Two opposite NUEFs were generated in the first part of the main channel by applying two different V_PP_ amplitudes on each side of the channel. These two NUEFs allowed the dielectric particles in the main channel to reach an equilibrium, resulting in particles being confined to the center of the channel. The equilibrium depended on the voltage amplitude and the frequencies of the power supply. Then, the third AC power supply was applied to perform the particle separation in the second part of the main channel. The suspension, containing polystyrene beads with a diameter of ≈ 5 μm and yeast cells, was used to evaluate the chip’s performance. This microfluidic chip, with this combination of castellated electrodes and an insulator, provides a good alternative option for cell separation by DEP. The combination of insulator sidewalls and castellated electrode achieved accurate particle focusing and continuous separation.

A microfluidic device, combining interdigitated and castellated electrodes, was developed to quantitatively assess the particle and cell separation (Figure 6C) [71]. The Pt electrodes were patterned to form both an interdigitated electrode system as well as castellated topology on a glass substrate. The height, width, and gap of the interdigitated electrodes and castellated electrodes were ≈ 100 nm × 10 μm × 10 μm and ≈ 100 nm × 20 μm × 20 μm, respectively. The fluidic chamber, with a length, width, and height of ≈ 10 mm, ≈ 4 mm, and ≈ 30 μm, respectively, was made of poly(methyl methacrylate) (PMMA), and then the PMMA was bonded to a glass substrate on top of the electrodes. The suspension of polystyrene beads with a diameter of ≈ 500 nm, ≈ 2 μm, and ≈ 6 μm, and RBCs were used to quantitatively evaluate the system’s performance. The suspension was pumped into the chamber, and an AC power supply was applied with a set value ranging from 15 V_PP_ to 20 V_PP_ and from 1 MHz to 15 MHz.

An AC-DEP chip was developed with a low-cost screen-printing thick-film technique (Figure 6D). A glass slide was used as a substrate, and a woven mesh and electrodes first patterned a commercial carbon paste, and contact wires were printed on its surface as using screen printing [63]. Three types of electrodes, ≈ 10 μm thick, were printed on the glass: offset castellated electrodes, non-offset castellated electrodes, and interdigitated line electrodes. Once the carbon paste hardened, the ultraviolet (UV) curable dielectric paste was then printed and cured using UV light to form microfluidic channels with a depth of ≈ 50 μm, and the contact wires were separated from the fluid. Finally, the PDMS layer with inlet and outlet ports was bonded on top of the channels, assisted by an oxygen plasma surface-activation method. The device’s performance and efficiency in capturing yeast cells from the suspension were optimized using an AC electric field of different frequencies and strengths. The result indicated that the voltage amplitude, the frequency of AC power supply, the carrier fluid conductivity, and the flow rate significantly affected the capture rate. The capture rate was > 95% after applying the AC power supply with set values of 10 V_PP_ at 100 kHz. This simple fabrication technique has great potential to be applied in mass production.

#### 4.2.3. Other Types of Electrodes

In addition to the main types of electrode shapes, many different electrodes with specific shapes were designed to provide better and more stable NUEF, such as annular [72,73], oblique [74,75,76], and curved electrodes [77]. The characteristics of these electrodes have been reviewed elsewhere [51] and are not the subject of this study; here, we only introduce the quadrupole and trapezoid electrodes.

The performance of quadrupole electrodes was first simulated, designed, and fabricated based on a polynomial with the assumption that it obeyed Laplace’s equation [66] to generate a well- defined NUEF that separates yeast cells. A similar electrode type was then used to pattern actin using DEP on a myosin substrate along electric field lines.

A microfluidic platform was proposed to separate bacteria from whole blood via DEP generated by a quadrupole microelectrode embedded at the bottom of a small well in the chip [65] (Figure 7). The blood was pumped into the device and first desalinated by inline dialysis without membrane. Then, the target bacteria were separated and concentrated using DEP. This obtained a concentration efficiency of ≈ 79% and ≈ 78% of *Escherichia coli* and *Staphylococcus aureus* at a processing rate of ≈ 0.6 mL·h^−1^. The results were verified by performing a polymerase chain reaction, showing that bacterial 16S ribosomal DNA levels were enriched ≈ 307-fold in comparison with the original.

A trapezoidal electrode array (TEA) based microfluidic chip was developed to separate polystyrene beads of different diameters (Figure 8) [78]. The gold TEA was patterned on a Pyrex glass substrate using a photolithography method. The widths of two bases in a single trapezoidal electrode were ≈ 120 μm and ≈ 20 μm, respectively, and their height was ≈ 60 μm. The microfluidic structure was patterned in PDMS using a soft lithography method. The width and depth of the microfluidic channel were ≈ 50 μm and ≈ 30 μm, respectively. The fluidic structure was aligned to the TEA, and then the PDMS with a patterned structure was bonded to the glass substrate using an oxygen plasma method.

The system had three sections: focus, separation, and collection. In the focus section, 100 trapezoidal electrodes forced the flow that contained polystyrene beads to move in a streamline along one side of the channel. In the separation section, a TEA with 20 electrodes performed the bead separation based on the beads’ diameter, achieving different dielectrophoretic velocities. The streams of the two separated beads were split into two outlets and collected in the collecting section.

The polystyrene beads with diameters of ≈ 6 μm and ≈ 15 μm were used to test the separation efficiency by applying an AC power supply with a set value of 8 V_PP_ at 50 kHz. The effect of the flow rate was investigated to evaluate the separation performance. The results showed that the purities of beads with diameters of ≈ 6 μm and ≈ 15 μm were 96.8 ± 0.6% and 99.5 ± 0.5%, respectively, at a flow rate of ≈ 10 mL·h^−1^. In comparison with other DEP devices, the TEA system gently processed the sample to prevent cell damage from long-term exposure in the electric field.

The integrated circuit (IC) technique was used to fabricate the electrode array to manipulate yeast cells and mammalian cells (Figure 9) [79]. The IC electrodes array with a static random access memory element was designed and fabricated using a standard complementary metal-oxide-semiconductor (CMOS) process. The array, with a size of ≈ 1.4 mm × 2.8 mm, consisted of 32,768 individual pixels, and each pixel was ≈ 11 µm × 11 µm. The microfluidic channel was formed in a double-sided adhesive film placed on the IC electrode array and then sealed by a coverslip with holes connected to the fluid input and output tubes. The entire setup was placed on a Cu block serving as a cooler. The DEP was performed in a set value of 5 V_PP_ at 1.8 MHz to test the cell manipulation, resulting in cell movement at a speed of ≈ 30 µm s^−^^1^. Additionally, this setup could split the water from oil in a volume of pL. This IC microfluidic system has good potential to be widely used because the electrode array can be mass produced using a standard CMOS fabrication process.

### 4.3. Three-Dimensional Electrodes

We discussed the DEP microfluidic chips with a variety of 2D electrodes. However, *F*_DEP_ is mainly available at the bottom of the microfluidic channels near the thin-film electrodes. There is a significant decrease of *F*_DEP_ in a vertical direction [64], which is inversely proportional to the distance from the electrodes generating the electric field. The polarized particles that flow at a certain distance above the electrodes might not be dielectrophoretically trapped leading to the development of a 3D electrode system to perform a more powerful NUEF through the entire depth of the channels. This also provided a gradient in the velocity generated by the hydrodynamic force in the device, improving the separation performance of the microfluidic devices [80,81,82,83,84,85,86,87,88,89,90].

#### 4.3.1. Metal 3D Electrodes

The thin-film electrodes patterned on the side wall of the main channel could be used as 3D electrodes (Figure 10) [28]. Compared to the planar electrodes at the bottom of the channel, the electrodes patterned on both side walls provided a much stronger NUEF and suppressed the field strength decay along the channel’s height, resulting in cell focusing and separation [70]. The kidney cells HEK293/nerve cells N115 and kidney cell HEK293/polystyrene beads were used to verify the chip’s performance by applying an AC power supply with optimized voltage and frequency.

A microfluidic device made by PDMS molding, combining acoustophoresis and DEP methods, was proposed, designed, fabricated, and tested to wash and separate uncoated and half-coated latex particles with aluminum with a mean diameter of ≈ 5 µm (Figure 11A,B) [80]. The brass mold was machined, making microfluidic channels with a depth and length of ≈ 200 μm and ≈ 61 mm, respectively. Two piezoelectric ceramic transducer slides made of lead-zirconate-titanate (PZT) and metal electrodes were placed next to each other and tightened at both sides of the channels on the brass mold. The liquid PDMS precursor was then poured into the mold and cured. Finally, the cured PDMS, patterned with structures attached to PZTs and electrodes, was peeled off from the mold and bonded to a glass slide using an oxygen plasma surface-activation process. The chip consisted of two main units, one for washing and one for separation. The suspension containing target particles, with contents of ≈ 1 × 10^7^ particles·mL^−1^, and the buffer were pumped into the channel from two different inlets. The standing waves generated by the PZTs in the washing section of the chip caused the particles to move from the carrier into the buffer solution with low electrical conductivity, effectively suppressing the amplitude of the Joule heating effect. Subsequently, the solution reached the 3D separation unit of the chip, where the *F*_DEP_ separated the uncoated and half-coated particles using an AC power supply with set values of 16 V_PP_ at 3 MHz. Integrating the washing and separation units provided a valuable system for cell separation in clinical diagnostics. However, the fabrication of the 3D electrodes is a challenging process. Additionally, the process of tightening the screws into the brass mold is another challenge because there is a high chance of liquid leakage.

#### 4.3.2. Carbon 3D Electrodes

Apart from the metal electrodes, electrodes made of carbon were also used in DEP microfluidic devices as carbon is more electrochemically stable than metal, resulting in the application of a higher voltage without electrolysis of the solution. The cost of the material and process was also lower than for metal electrodes as carbon electrodes were formed directly from the photoresist. These advantages are of interest to researchers developing carbon electrodes to perform DEP in microfluidics.

A 3D carbon electrode system was proposed to concentrate λ-DNA (Figure 11C) [81]. The carbon electrodes were made by carbonization of SU-8 photosensitive material. The fused silica substrate was coated with SU-8, then the SU-8 was patterned, exposed to a temperature of ≈ 900 °C, and carbonized. These carbon electrode pillars had a height and diameter of ≈ 95 μm and ≈ 58 μm, respectively. They were then connected to the carbon pads by dispensing indium. The microfluidic channels, with a depth and width of ≈ 100 μm and ≈ 2 mm, respectively, were made on a stack of pressure-sensitive double-sided adhesive tape with a thickness of ≈ 100 μm and then covered by a layer of polycarbonate (PC).

Finally, both substrates, fused silica with carbon electrodes and the stack of PC layers, were adhered to each other and sealed using a rolling press. A suspension containing λ-DNA was pumped into the channel with a flow rate of ≈ 2 μL·min^−1^. The AC power supply amplitude was set to 16 V_PP_ with different frequencies to characterize the system’s performance. The result showed that λ-DNA was trapped by pDEP or nDEP at set frequencies between either 10 kHz or 50 kHz for pDEP and either 75 kHz or 250 kHz for nDEP.

A compact disc (CD) microfluidic system with 3D carbon electrodes was developed to separate latex particles, with a diameter of ≈ 8 µm, from yeast cells (Figure 12) [91]. The microfluidic system consisted of four sections: a programmable motorized spin stand combined with an optical detection system, a function generator to apply the AC power supply, a CD platform to hold the DEP chips when spinning, and carbon-DEP chips. The formation of carbon electrodes was described in [81]. The SU-8 was first patterned on the substrate, and then the pyrolysis was conducted to carbonize the SU-8, forming the 3D electrodes at a height of ≈ 40 µm and ≈ 70 µm, respectively. All the electrodes were connected to the electric leads. A thin SU-8 layer was patterned around the 3D electrodes to protect the connecting leads from contact with the suspension.

The fluidic structure with two chambers and two channels was patterned in PC and sealed by a double-sided pressure-sensitive adhesive. The width and height of the main channel were ≈ 600 µm and ≈ 100 µm, respectively. The width and height of the channel for suspension retrieval were ≈ 1 µm and ≈ 100 µm, respectively. The loaded suspension in the inlet chamber was delivered into the main channel, and its flow rate was controlled by a centrifugal force caused by rotating the CD at ≈ 800 revolutions per minute. The AC power supply with a set value of 20 V_PP_ at 200 kHz was applied to separate the yeast cells from latex particles at a flow rate of ≈ 35 mL·min^−1^.

#### 4.3.3. Polymer 3D Electrodes

The mixture of silver platelet PDMS was another alternative to fabricate the electrode. Different from the fabrication of a metal electrode, it removed the sputtering and photolithography applied in metal film electrodes, simplifying the fabrication process and device structure.

The 3D electrodes were developed based on PDMS (Figure 13) [89]. The ITO leadouts were first patterned on a glass substrate using a photolithography method. A second photolithography was then applied to form the 3D electrode mold. Ag-PDMS 3D electrodes made of silver platelets with a diameter of ≈ 1 µm and PDMS precursor mixture with a ratio of 85% were poured on the top of the ITO leadouts and the PDMS precursor was subsequently cross-linked. The width and gap of the electrodes were both ≈ 200 µm. The PDMS layer patterned with inlets, outlets, and a channel made by soft lithography was aligned to the electrodes and bonded to the glass substrate using an oxygen plasma activation method. The diameters of the two inlets and outlets were ≈ 6 mm and ≈ 5 mm, respectively. The width of the channels connected to inlet A and inlet B was ≈ 100 µm and ≈ 200 µm, respectively. The width of the main channel was ≈ 200 µm. There were three vaulted obstacles along the main channel with a pitch and radius of ≈ 400 µm and ≈ 100 µm, respectively.

The suspension containing yeast cells and gold-coated polystyrene microspheres with a diameter of ≈ 25 µm was pumped into the channel connected to inlet A. The suspension was moving along the upper side wall of the main channel by the dynamic focusing flow generated by the buffer from the channel connected to inlet B. The AC power supply was applied with different amplitudes of V_PP_ at 1 MHz to investigate the separation efficiency at different flow rates. The results showed that > 90% yeast cells and ≈ 100% gold-coated polystyrene microspheres were separated by applying voltages of ≈ 18.75 V_PP_ and ≈ 12.5 V_PP_ at 1 MHz with a flow rate of ≈ 300 µm·s^−^^1^.

A 3D Ag-PDMS electrode microfluidic device was developed to separate microalgae cells in ballast water (Figure 14) [90]. The electrode fabrication process was described in [89]. The length and width of the main channel were ≈ 2000 µm and ≈ 100 µm, respectively. One side of the main channel was designed as a triangular shape to increase the inhomogeneity of the electric field and improve the *F*_DEP_. The suspension containing cells of interest and the buffer were pumped into the channel from two inlets forcing the cells to move along the triangular side of the main channel. Two types of microalgae cells, *Platymonas* and *Closterium*, and polystyrene particles were used to test the chip’s performance by applying an optimized AC power supply with a value of ≈ 10 V_PP_ at 30 MHz, obtaining a separation efficiency of > 90%.

#### 4.3.4. Silicon 3D Electrodes

The developments in IC and micro-electro-mechanical-system fabrication technology have also been used for DEP, allowing the chip structure to be made from single crystal silicon. As a result, the chip fabrication and structure were simplified as using electrically conductive silicon formed both, the microfluidic channel walls also serving as 3D electrodes, obtaining a potential for mass production.

A square-shaped pillar structure together with the channels was proposed and fabricated from silicon (Figure 15A) [82] using a deep reactive ion etching (DRIE) process and was then sandwiched in silicon with two glass wafers by anodic bonding. Living/dead yeast cells were used to test the device’s performance by applying the AC power supply with a gradually increasing amplitude from 0 V_PP_ to 25 V_PP_ at a frequency range from 20 kHz to 100 kHz.

The electrode pillars could also be designed in a non-uniform height to separate living and dead yeast cells (Figure 15B) [83]. The device consisted of two glass wafers sandwiching a patterned silicon wafer forming the 3D electrodes as well as the separation chamber. The glass wafer was anodically bonded to the silicon wafer, and then the DRIE was performed to etch the silicon, forming two sets of pillars of different heights, either ≈ 100 μm or ≈ 2.5 μm, as well as the chamber. These two structures were then bonded together, forming the asymmetric electrode structures. The device’s performance was tested using two populations of cells, dead and live yeast, one exhibiting the pDEP and the other exhibiting nDEP, using an AC power supply with a set amplitude of 20 V_PP_ at 20 kHz.

Another DEP microfluidic device with symmetric silicon electrodes was proposed to perform a sequential DEP field flow cell separation (Figure 15C) [84]. The chip fabrication process was similar to the previous one shown in Figure 15B. The only difference was that the silicon electrodes were symmetrical. This electrode geometry, compared to the coplanar electrodes, suppressed Joule heat, and increased the separation efficiency. The suspension was first pumped into the channel, and the particles were then trapped in different locations by performing the DEP. One population of particles (group A) was trapped in the center of the channel, which was the weakest position of the electric field, while the other population (group B) was trapped near the electrode field. Group A was then flushed out of the channel by increasing the flow rate. The electric field was then removed, and group B was subsequently flushed out. This array structure increased separation efficiency. Living/dead yeast cells were used to verify the chip’s performance.

Silicon can also be used as a bulk electrode without arrays, simplifying the chip structure and fabrication process (Figure 15D) [85]. The simulation showed that there was no volume without a NUEF, suggesting good separation efficiency. The fabricated chip consisted of three parts: heavily doped silicon patterned with corrugated channels forming the DEP, sandwiched by two glasses mechanically connected by two subsequent anodic bonding processes. The device’s performance was again verified by separating dead from living yeast cells using an AC power supply with a set voltage amplitude of 25 V_PP_.

## 5. Conclusions

DEP is a particle separation technique based on particle polarization in a NUEF. It has been proven to be a viable tool for LOC in the application of medicine, molecular diagnostics, and any other fields related to particle separation because of its selectivity and accuracy. It is also a label-free method. In this review, we described DEP theory and compared it to other separation techniques to show its advantages for microfluidic devices. We also introduced the recent developments in DEP microfluidic devices, classified by electrode type. For each system, we explained the chip structure, fabrication process, electrode type, and operation principle. These microfluidic devices have the potential to be widely used in laboratories and diagnostic centers.

However, the wider employment of DEP would probably require higher levels of system integration, including sample pretreatment, post DEP processing, as well as product analysis. In addition, the DEP itself should be designed in a more efficient way of using 3D or semi-3D operations. That could be achieved either by designing the 3D electrode geometry or making the DEP area shallow and suppressing the low efficiency space to promote the commercialization and wide utilization of DEP.

## Figures and Tables

**Figure 1 micromachines-10-00423-f001:**
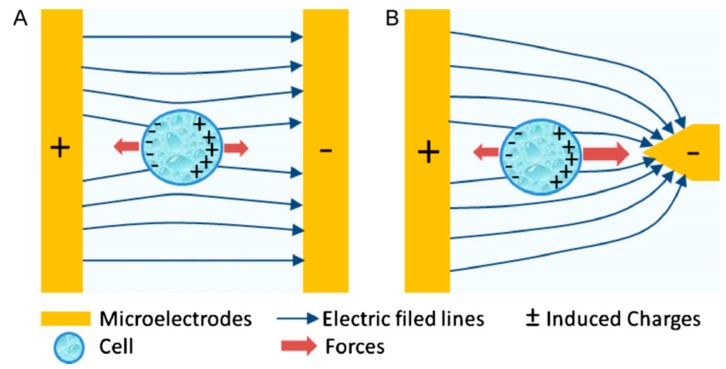
Diagram of the positive dielectrophoresis (pDEP) principle. Reproduced with permission from [51], © 2011 Elsevier. (**A**) The particle is symmetrically polarized in a uniform electrical field, resulting in net DEP force (*F*_DEP_) with zero amplitude. (**B**) The particle is asymmetrically polarized in the non-uniform electric field (NUEF) and the net *F*_DEP_, resulting in moving the particle into an area with the maximum amplitude of the electric field.

**Figure 2 micromachines-10-00423-f002:**
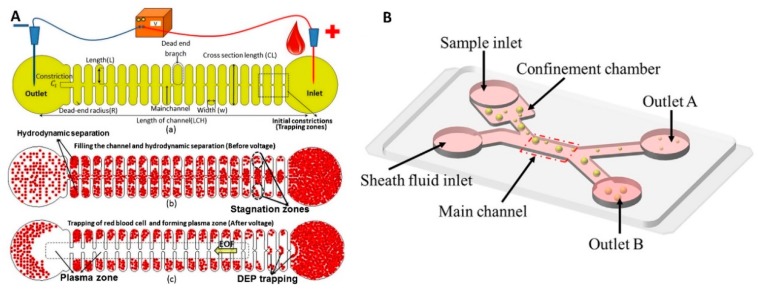
DEP devices with external electrodes. (**A**) Schematic of iDEP chips achieving plasma separation without diluting the blood samples. Reproduced with permission from [54], © 2015 Springer Nature. (**B**) Schematic of chip used for heterogeneous emulsion droplets separation. Reproduced with permission from [56], © 2017 Elsevier.

**Figure 3 micromachines-10-00423-f003:**
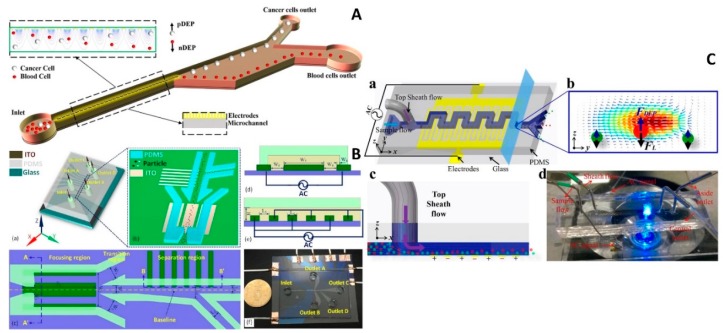
2D electrode microfluidic chips with DEP. (**A**) Schematic of the device to separate cancer cells from blood cells by *F*_DEP_ generated by parallel electrodes. Reproduced with permission from [52], © 2016 John Wiley and Sons. (**B**) The device combined with induced-charge electro-osmosis (ICEO) for particle focusing and DEP achieving simpler chip structure to perform particle focusing and separation. Reproduced with permission from [60], © 2017 American Chemical Society. (**C**) Hybrid DEP-inertial device with top sheath flow to improve the separation efficiency. Reproduced with permission from [64], © 2018 Elsevier.

**Figure 4 micromachines-10-00423-f004:**
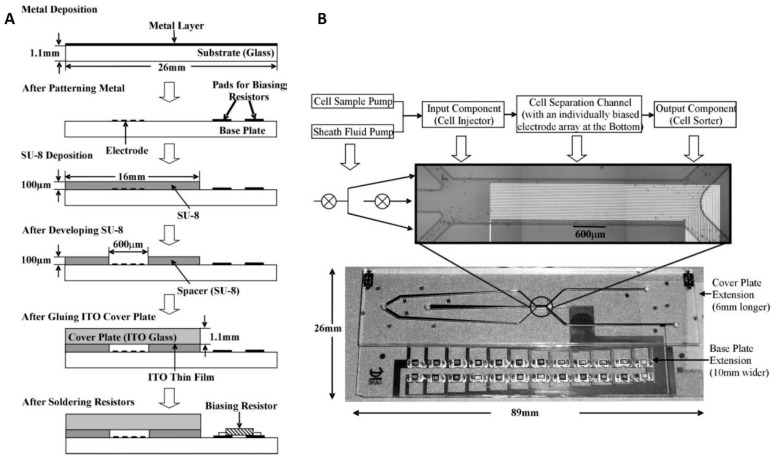
An isomotive DEP microfluidic device with parallel electrodes. (**A**) The fabrication process of the device. (**B**) Diagram of sample process sequence and photograph of the device with details showing the main channel for DEP. Reproduced with permission from [67], © 2007 Royal Society of Chemistry (RSC).

**Figure 5 micromachines-10-00423-f005:**
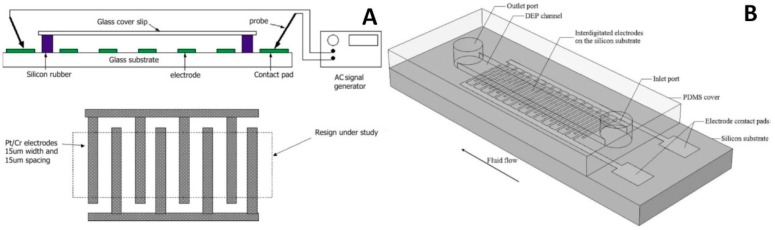
Devices with interdigitated electrodes. (**A**) Schematic of the system with simple particle manipulation and top view of electrodes. Reproduced with permission from [68], © 2002 Elsevier. (**B**) Drawing of the chip combining antibody recognition with DEP, thus improving the selectively of capturing bacteria. Reproduced with permission from [69], © 2006 RSC.

**Figure 6 micromachines-10-00423-f006:**
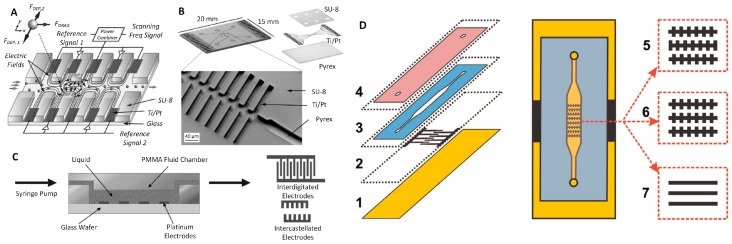
Devices with castellated electrodes. (**A**) Schematic of an insulator and 2D electrode device to focus and separate particles by applying different frequencies and amplitudes of an AC power supply [70]. (**B**) 3D drawing and scanning electronic microscopy image of the device. Reproduced with permission from [70], © 2008 Elsevier. (**C**) Schematic of the combined interdigitated and castellated electrode device. It was used to quantitatively evaluate the particle and cell separation. Reproduced with permission from [71], © 2003 Elsevier. (**D**) Schematic of a device based on screen-printing fabrication. Reproduced with permission from [63], © 2015 Elsevier.

**Figure 7 micromachines-10-00423-f007:**
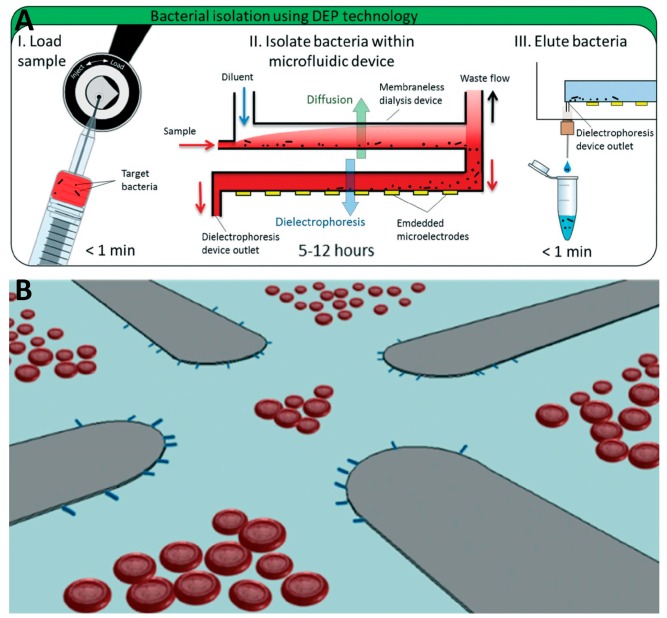
A device with quadrupole electrodes. (**A**) The principle of the device. (**B**) Schematic showing red blood cells (RBC) separation by quadrupole electrode configuration. Reproduced with permission from [65], © 2017 RSC.

**Figure 8 micromachines-10-00423-f008:**
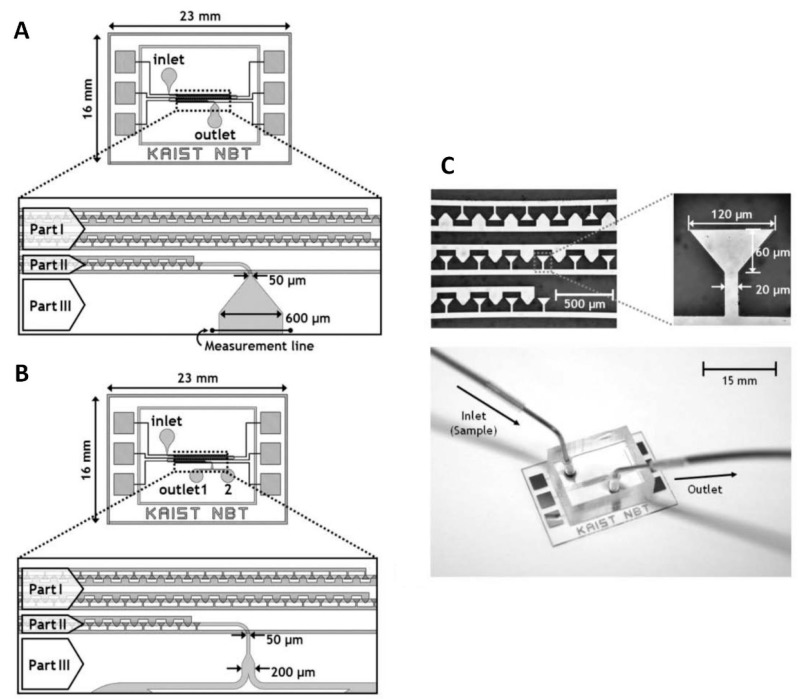
A trapezoidal electrode array (TEA) microfluidic chip. Reproduced with permission from [78], © 2005 RSC. (**A**) Schematic of the chip for particle separation. (**B**) Schematic of the chip for particle fractionation. (**C**) Image of the TEA and the device.

**Figure 9 micromachines-10-00423-f009:**
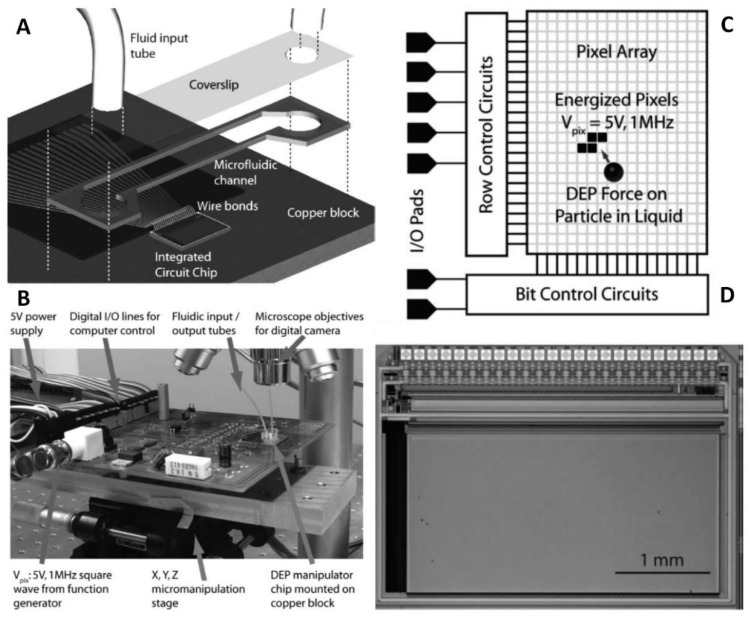
An integrated circuit (IC) DEP microfluidic system. Reproduced with permission from [79], © 2008 RSC. (**A**) Schematic of the device. (**B**) Photograph of the system. (**C**) The schematic of the IC electrodes. (**D**) Image of the IC electrodes.

**Figure 10 micromachines-10-00423-f010:**
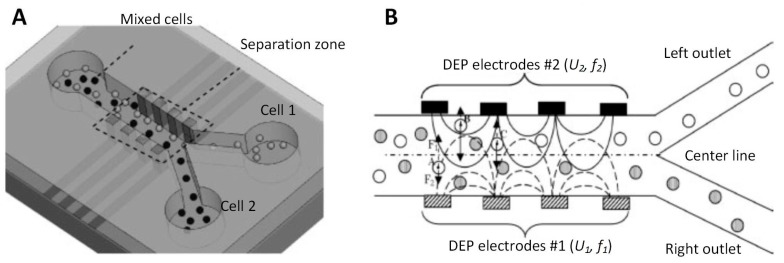
A vertical thin-film device and the 2D thin-film electrodes were used as 3D electrodes. Reproduced with permission from [28], © 2009 John Wiley and Sons. (**A**) Schematic of the device. (**B**) The drawing of the cell separation in the separation zone.

**Figure 11 micromachines-10-00423-f011:**
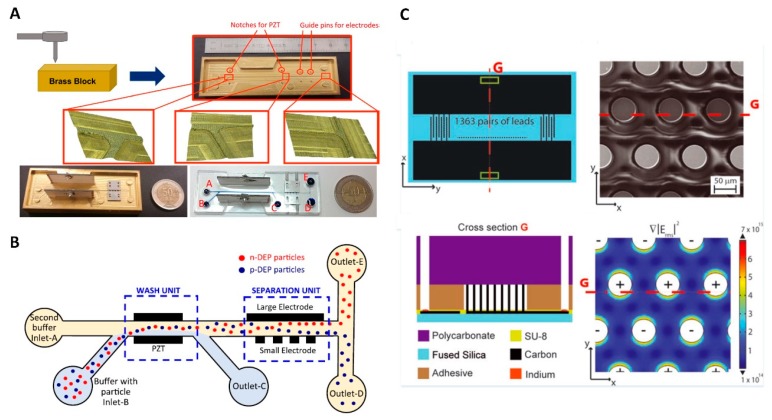
Devices with 3D electrodes. (**A**) Photograph of the device combined with piezoelectric ceramic transducer slides made of lead-zirconate-titanate (PZT) slides and metal electrodes. (**B**) Principle of the device. The PZT slides were used to focus the particles, and the particles were separated in the following NUEF area. Reproduced with permission from [80], © 2016 American Institute of Physics (AIP) Publishing. (**C**) Schematic and simulation of the device with a carbon electrode array. Reproduced with permission from [81], © 2013 John Wiley and Sons.

**Figure 12 micromachines-10-00423-f012:**
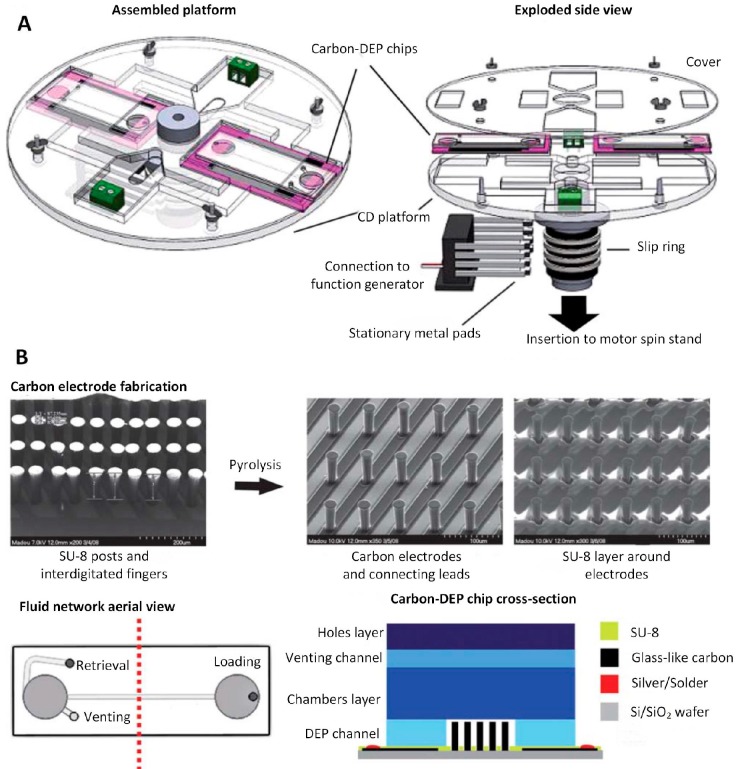
Compact disc (CD) microfluidic device with 3D carbon electrodes. Reproduced with permission from [91], © 2010 RSC. (**A**) Schematic of the device. (**B**) The image of the carbon electrodes and the schematic of the chip.

**Figure 13 micromachines-10-00423-f013:**
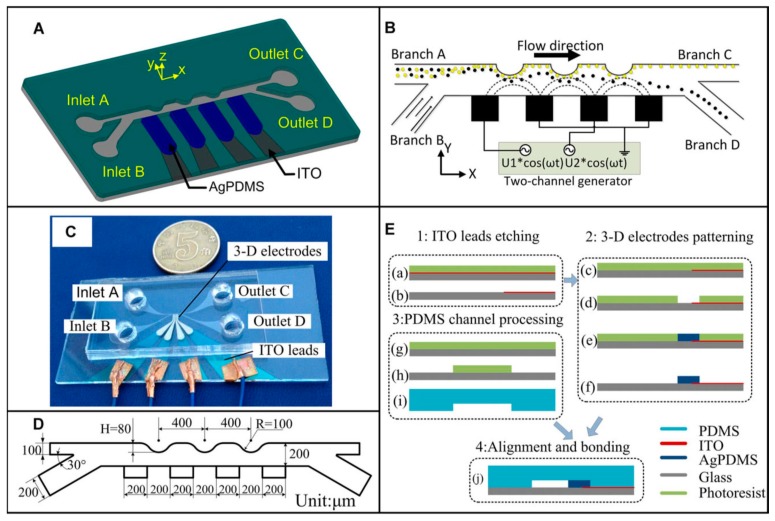
A 3D Ag- polydimethylsiloxane (PDMS) electrode microfluidic device. Reproduced with permission from [89], © 2015 John Wiley and Sons. (**A**) Schematic of the device. (**B**) The drawing of the separation. (**C**) The photo of the device. (**D**) The dimension of the structure. (**E**) The fabrication process of the device.

**Figure 14 micromachines-10-00423-f014:**
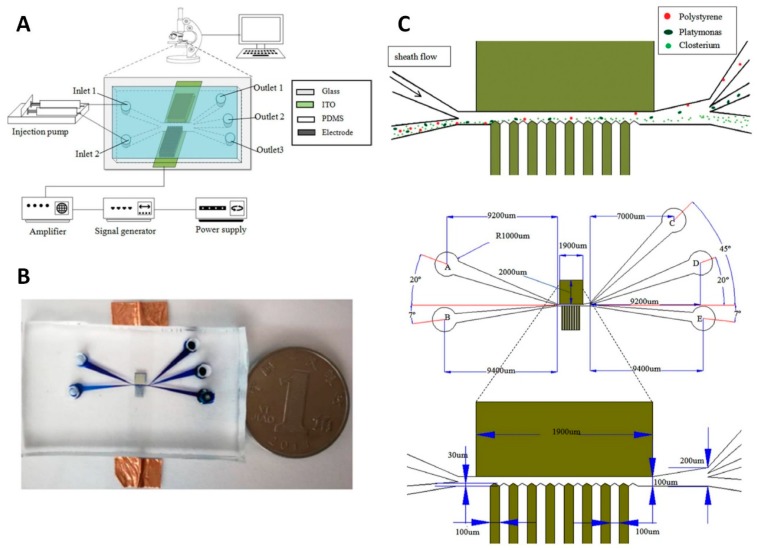
A 3D Ag-PDMS electrode microfluidic device. Reproduced with permission from [90], © 2019 John Wiley and Sons. (**A**) Schematic of the system. (**B**) The image of the device. (**C**) The schematic in the separation area and the dimension of the electrodes.

**Figure 15 micromachines-10-00423-f015:**
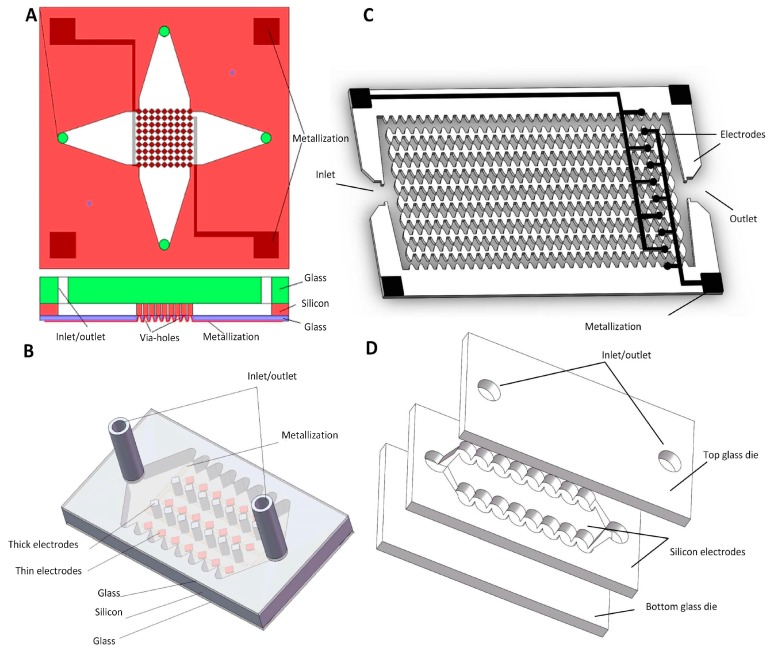
Microfluidic devices with silicon 3D electrodes. (**A**) Schematic of the device with silicon electrode pillars of a uniform height. Reproduced with permission from [82], © 2008 Elsevier. (**B**) Schematic of the device with silicon electrode pillars of a non-uniform height. Reproduced with permission from [83], © 2009 AIP Publishing. (**C**) Schematic of the device with symmetrical silicon electrodes [84]. (**D**) Schematic of the device with bulk silicon electrode [85].

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
