# Peer review of "DEP-on-a-Chip: Dielectrophoresis Applied to Microfluidic Platforms"

_micromachines, 2019, doi:10.3390/mi10060423_

Round 1
Reviewer 1 Report
Zhang et al. present a review of dielectrophoresis. They mainly describe electrode based dielectrophoresis and present several devices and their specifications and the respective fabrication methods. Unfortunately they lack clarification which readers they want to address. Additionally, a review of dielectrophoresis should at least provide the basic theory of the physical effect and the main equations describing the forces, this is completely lacking in the manuscript. It is questionable what the present review offers as new aspect that is an asset to previous reviews about dielectrophoresis (like Jubery et al., Electrophoresis, 2014; Viefhues and Eichhorn, Electrophoresis, 2017; Xuan, Electrophoresis, 2019; Kim et al. Anal Chem, 2019). Thus, the manuscript is not recommended for publication in the present form. See also the following points:
· ll 30-38: The reviewer wonders why the authors referred to specific research papers in the introduction rather than providing respective review paper of the different processes.
· ll 59 ff: The authors listed some techniques for particle separation. What are the respective limits of the techniques, e.g. size resolution, time, sample loss etc.? Optical tweezers are not used for separation purposes to the reviewer’s knowledge and thus should not be listed in that context.
· ll 90 ff: in a uniform electric field there is no dielectric force. For instance, it is not balanced, it is just zero.
· ll 96 ff: this paragraph is unclear.
· A critical discussion of the techniques and devices used in the papers is lacking. The authors describe various devices that exploit DEP for different applications. They should add advantages and disadvantages, especially of the different types of 2D electrode devices to provide the reader a guide through the existing ones.
· It is unclear according to which scheme the different electrode types were divided in the sections. For example, why are parallel and interdigitated electrodes in different sections? What is the main aspect that affects the performance?
· Figure 3 c and d): why are they listed here? They are mentioned in the next section, see also previous comment.
· iDEP: why are the authors limiting the iDEP to DC DEP? There have been shown several applications that were due to AC DEP. Additionally, it remains completely unclear how the inhomogeneous electric field was generated, e.g. by insulating posts, hurdles, tapers, curved channels, etc. iDEP offers other advantages and disadvantages than DEP based on microelectrodes, see for example Regtmeier et al. Electrophoresis, 2011. The authors should discuss them.
· Section 4.3 is quite long. It is recommended that the section is subdivided. So far, the reader is overwhelmed by the large number of different devices. Those should be clustered sufficiently, e.g. by their fabrication or functionality.
Author Response
Zhang et al. present a review of dielectrophoresis. They mainly describe electrode based dielectrophoresis and present several devices and their specifications and the respective fabrication methods. Unfortunately, they lack clarification which readers want to address. Additionally, a review of dielectrophoresis should at least provide the basic theory of the physical effect and the main equations describing the forces, this is completely lacking in the manuscript.
The theory and equations were added at theory section and a very short clarification was added at lines from 43 to 45.
It is questionable what the present review offers as new aspect that is an asset to previous reviews about dielectrophoresis (like Jubery et al., Electrophoresis, 2014; Viefhues and Eichhorn, Electrophoresis, 2017; Xuan, Electrophoresis, 2019; Kim et al. Anal Chem, 2019). Thus, the manuscript is not recommended for publication in the present form. See also the following points:
The related texts are in line 42-45 and we also added some texts to show the purpose of the review.
· ll 30-38: The reviewer wonders why the authors referred to specific research papers in the introduction rather than providing respective review paper of the different processes.
The references of review papers were added into introduction section
· ll 59 ff: The authors listed some techniques for particle separation. What are the respective limits of the techniques, e.g. size resolution, time, sample loss etc.? Optical tweezers are not used for separation purposes to the reviewer’s knowledge and thus should not be listed in that context.
Here we only considered applications suitable for microfluidics. We cannot list the specific limits for each method as there are too many materials to perform these methods and the parameters are different from each other.
The description of optical tweezers was deleted.
· line 90: in a uniform electric field there is no dielectric force. For instance, it is not balanced, it is just zero.
It sentence was modified accordingly.
· ll 96 ff: this paragraph is unclear.
The text was modified and we added short description of DEP and the fundamental governing DEP equation.
· A critical discussion of the techniques and devices used in the papers is lacking. The authors describe various devices that exploit DEP for different applications. They should add advantages and disadvantages, especially of the different types of 2D electrode devices to provide the reader a guide through the existing ones.
We added the brief advantages of different types of 2D electrodes into each section
· It is unclear according to which scheme the different electrode types were divided in the sections. For example, why are parallel and interdigitated electrodes in different sections? What is the main aspect that affects the performance?
We reorganized the section: we put the parallel and interdigitated electrodes into one section and briefly explained the performance of the electrodes.
· Figure 3 c and d): why are they listed here? They are mentioned in the next section, see also previous comment.
The figures are reorganized.
· iDEP: why are the authors limiting the iDEP to DC DEP? There have been shown several applications that were due to AC DEP. Additionally, it remains completely unclear how the inhomogeneous electric field was generated, e.g. by insulating posts, hurdles, tapers, curved channels, etc. iDEP offers other advantages and disadvantages than DEP based on microelectrodes, see for example Regtmeier et al. Electrophoresis, 2011. The authors should discuss them.
We modified the sentence accordingly and we also added the above mentioned reference.
· Section 4.3 is quite long. It is recommended that the section is subdivided. So far, the reader is overwhelmed by the large number of different devices. Those should be clustered sufficiently, e.g. by their fabrication or functionality.
The section was split into subsection by adding subtitles.
Reviewer 2 Report
Review on: DEP-on-a-chip: Dielectrophoresis applied in microfluidic platforms, by Haoqing Zhang, Honglong Chang and Pavel Neuzil.
The review is very well written. It is highly appreciated that most examples are very precisely illustrated. Not less then 15 figures are given.
Only 88 references are given but they correctly cover the field, which is developing.
Minor comments
Introduction
Redundancy between lines 30-35 and 48-50.
Line 88: “In spite of electrophoresis…” may be replaced by “Conversely to…”
Major comments
- For every example cited from the literature, the authors give many fabrication details (thicknesses, etc). For a reader wanting o precisely known how the devices are made, it could be of interest (but it is then easy to read the original article); for the reader of this review, it is long, not focused to the point of interest of each device, and therefore does help to make the review clear. Instead of that, the authors should insist on the originalities of each example.
- Related to the previous comment: the authors are invited to point out, for each example, which point is original compared to other DEP devices. For example, what are the difference, insisting in performances, the parallel electrodes, 2D, interdigitated or castellated (or annular, quadrupole, etc.), 3D (pillars, etc.).
- The conclusion is not a conclusion but a quick summary. My comments are the same as above: the review should be more critical and should propose a trend: what researchers are expected to do now on DEP.
My conclusions: The authors should make their review more critical; it is expected to bring new information out of the comparison of all published work, not only a (precise, in this case) description of what has been done until now.
Author Response
Review on: DEP-on-a-chip: Dielectrophoresis applied in microfluidic platforms, by Haoqing Zhang, Honglong Chang and Pavel Neuzil.
The review is very well written. It is highly appreciated that most examples are very precisely illustrated. Not less than 15 figures are given.
Only 88 references are given but they correctly cover the field, which is developing.
Minor comments
Introduction
Redundancy between lines 30-35 and 48-50.
The text of 48-50 has been deleted.
Line 88: “In spite of electrophoresis…” may be replaced by “Conversely to…”
The sentence was deleted.
Major comments
- For every example cited from the literature, the authors give many fabrication details (thicknesses, etc). For a reader wanting to precisely known how the devices are made, it could be of interest (but it is then easy to read the original article); for the reader of this review, it is long, not focused to the point of interest of each device, and therefore does help to make the review clear. Instead of that, the authors should insist on the originalities of each example.
We added short comments about the originalities of some examples where it has not been already written.
- Related to the previous comment: the authors are invited to point out, for each example, which point is original compared to other DEP devices. For example, what are the difference, insisting in performances, the parallel electrodes, 2D, interdigitated or castellated (or annular, quadrupole, etc.), 3D (pillars, etc.).
A brief discussion about this was added under each subtitle.
- The conclusion is not a conclusion but a quick summary. My comments are the same as above: the review should be more critical and should propose a trend: what researchers are expected to do now on DEP.
My conclusions: The authors should make their review more critical; it is expected to bring new information out of the comparison of all published work, not only a (precise, in this case) description of what has been done until now.
We did add a few comments to reflect the note above.
Reviewer 3 Report
General Comments
Each chip design is described in very careful detail but all the detail drowns out the author’s interpretation and their assignment of the work’s importance. The value of the review is not in a description of all the fine level details (such as specific dimensions). The value is the author’s assessment of the field and how these different technologies are pushing the value of the technique forward and what needs to be done to make advancements in the future. Instead of simply summarizing what is in each paper the authors should try to point out the important advance each design has made and connect that to the overall theme of the review which seems to be that DEP technology is progressing to improve particle separation under a variety of conditions. It’s not the summary that is important here but rather the author’s interpretation of the different papers and how the authors feel this fits into the larger applications for the technology.
Conclusions – The authors need to tie together all the information they presented not just as a summary of the chips, but also as their interpretation of where the field is going and what specific advances were made and what advances are needed to take DEP based separation to the next level.
Specific Comments
Line 27 – The majority of these chips are much larger than 1mm. They might have feature sizes that are smaller than 1mm, but the chips themselves are usually on the cm scale. This statement needs to be revised
Line 48-50 - No need to repeat all the different techniques already listed above.
Line 54 – Filtration is actually a very effective purification method. You need to emphasize in this first sentence that it is the use of filtration in a microfluidic chip that can have challenges.
Line 59 – Centrifugation based separation is really based on relative specific mass and not on the total weight. Two particles of equal density should sit in the same separation layer regardless of overall weight.
Line 62 – It’s really based on the magnetic susceptibility of the particle, not magnetism.
Line 69 – What do you mean by a virtual tool? It is a real tool that can move real objects
Line 88- It’s not in spite of, it’s simply different
Line 90 – It would be good to see a more thorough explanation of origin of the DEP force
Line 113 – The word “introduce” should be replaced with “discuss”
Figure 3 needs to be reorganized to be less visually confusing and needs to enlarge the figures so the details can be seen. The caption for each panel should summarize what is important about the figure instead of simply saying what it is. The value of the review is in the author’s interpretation, not just the summary of the literature.
Figure 7 needs to be expanded to more easily seen
Author Response
General Comments
Each chip design is described in very careful detail but all the detail drowns out the author’s interpretation and their assignment of the work’s importance. The value of the review is not in a description of all the fine level details (such as specific dimensions). The value is the author’s assessment of the field and how these different technologies are pushing the value of the technique forward and what needs to be done to make advancements in the future. Instead of simply summarizing what is in each paper the authors should try to point out the important advance each design has made and connect that to the overall theme of the review which seems to be that DEP technology is progressing to improve particle separation under a variety of conditions. It’s not the summary that is important here but rather the author’s interpretation of the different papers and how the authors feel this fits into the larger applications for the technology.
Conclusions – The authors need to tie together all the information they presented not just as a summary of the chips, but also as their interpretation of where the field is going and what specific advances were made and what advances are needed to take DEP based separation to the next level.
For these two comments, I really have no idea about the big direction of the DEP development as I did not work on it for a long time.
Specific Comments
Line 27 – The majority of these chips are much larger than 1mm. They might have feature sizes that are smaller than 1mm, but the chips themselves are usually on the cm scale. This statement needs to be revised
The statement about the sized was implemented accordingly.
Line 48-50 - No need to repeat all the different techniques already listed above.
The comment about different techniques was removed.
Line 54 – Filtration is actually a very effective purification method. You need to emphasize in this first sentence that it is the use of filtration in a microfluidic chip that can have challenges.
The comment regarding filtration and difficulties of this technique were added accordingly.
Line 59 – Centrifugation based separation is really based on relative specific mass and not on the total weight. Two particles of equal density should sit in the same separation layer regardless of overall weight.
We would like to thank the reviewer for this comment. The “total mass” was replaced with “relative specific mass”.
Line 62 – It’s really based on the magnetic susceptibility of the particle, not magnetism.
The statement was corrected accordingly.
Line 69 – What do you mean by a virtual tool? It is a real tool that can move real objects
We meant that there is actual tool but “invisible” forces. Nevertheless, we removed the word “virtual” to make the manuscript less confusing.
Line 88- It’s not in spite of, it’s simply different
The sentence was removed.
Line 90 – It would be good to see a more thorough explanation of origin of the DEP force
The text was modified and we added theory and fundamental governing equation of DEP.
Line 113 – The word “introduce” should be replaced with “discuss”
The word was replaced accordingly.
Figure 3 needs to be reorganized to be less visually confusing and needs to enlarge the figures so the details can be seen. The caption for each panel should summarize what is important about the figure instead of simply saying what it is. The value of the review is in the author’s interpretation, not just the summary of the literature.
We added a specific comments regarding the devices into the caption.
Figure 7 needs to be expanded to more easily seen
The Figure 7 was expanded as suggested.
Round 2
Reviewer 1 Report
The authors answered to all points of the 3 reviewers and could improve the manuscript. However, there are still some points that have to be considered before publication. Thus, publication is recommended after major revision. See the following points:
· Why do the authors refer to several paper-based microfluidics in the introduction? The general concept is quite different to that of microfluidics in glass or polymers.
· Improvement of English writing is needed for several new parts of the manuscript.
· Ll 93: DEP also needs different components, e.g. power supplies, and alignment of the electrodes to each other and/or to the fluidic system is difficult. Thus, those points are no proper reason to not use chromatography. But sample loss due to unspecific adhesion to the large surface of the stationary phase and the need for labeling in chromatography, while in DEP those fortunately don’t apply.
· L 108: in which case does F_DEP has zero amplitude. Moreover, “amplitude of force” is an insufficient wording.
· Ll 167: the concept of iDEP is not clearly stated here. Additionally, both, ac and dc electric fields are used in iDEP application and the NUEF is not generated by the power supply, but rather due to insulating posts, channel tapering, etc. in the microfluidic channel, see e.g. Regtmeier et al, Electrophoresis, 2011. Exploiting AC iDEP provides the advantage that linear and non-linear electric effects can be controlled independently.
· L231: the NUEF is not applied by the electrodes. It was generated in the channel tapering.
· The reviewer is missing a discussion of the current challenges of the described devices. For example, in plane electrodes generate an electric field that is strongly dependent on the distance to the electrodes, i.e. sample that is flown with certain distance above the electrodes might not be dielectrophoretically trapped.
Author Response
· Why do the authors refer to several paper-based microfluidics in the introduction? The general concept is quite different to that of microfluidics in glass or polymers.
We replaced the references to paper-based microfludics with microfluidic-based systems for sample pre-treatment, controlled drug deliveryu and on-chip reactions.
· Improvement of English writing is needed for several new parts of the manuscript.
The manuscript was proof read by native English speaking person
· Ll 93: DEP also needs different components, e.g. power supplies, and alignment of the electrodes to each other and/or to the fluidic system is difficult. Thus, those points are no proper reason to not use chromatography. But sample loss due to unspecific adhesion to the large surface of the stationary phase and the need for labeling in chromatography, while in DEP those fortunately don’t apply.
We added the note regarding disadvantages of the chromotagraphy per the reviewer’s suggestion.
· L 108: in which case does F_DEP has zero amplitude. Moreover, “amplitude of force” is an insufficient wording.
Sentences were modified accordingly.
· Ll 167: the concept of iDEP is not clearly stated here. Additionally, both, ac and dc electric fields are used in iDEP application and the NUEF is not generated by the power supply, but rather due to insulating posts, channel tapering, etc. in the microfluidic channel, see e.g. Regtmeier et al, Electrophoresis, 2011. Exploiting AC iDEP provides the advantage that linear and non-linear electric effects can be controlled independently.
Sentences were modified as follows:
The NUEF powered by the direct current (DC) or alternating current (AC) is generated by geometry of insulating posts in combination with channel tapering, etc. in the microfluidic channel [10] also known as AC/DC-iDEP [53].
· L231: the NUEF is not applied by the electrodes. It was generated in the channel tapering.
Sentences were modified as follows:
The NUEF was generated by the channel tapering after inserting four Pt electrodes into the inlets and outlets.
· The reviewer is missing a discussion of the current challenges of the described devices. For example, in plane electrodes generate an electric field that is strongly dependent on the distance to the electrodes, i.e. sample that is flown with certain distance above the electrodes might not be dielectrophoretically trapped.
We discussed the DEP-based microfluidic chips with a variety of 2D electrodes. However, FDEP is mainly available at the bottom of the microfluidic channels near the thin-film electrodes. There is a significant decrease of FDEP in a vertical direction [64], which is inversely proportional to the distance from the electrodes generating the electric field. The polarized particles flown in certain distance above the electrodes might not be dielectrophoretically trapped leading to a development of a 3D electrode systems to perform a more powerful NUEF through the entire depth of the channels. This also provided a gradient in the velocity generated by the hydrodynamic force in the device, improving the separation performance of the microfluidic devices [80-90].
Reviewer 3 Report
Line 857 - "However, much effort still needs to be devoted to the development of DEP-based LOC systems to promote their commercialization and wide utilization."
There should be more discussion on what the authors think needs to be done to promote DEP commercialization and wide utilization. What specific types of development is needed to move things forward?
Author Response
Line 857 - "However, much effort still needs to be devoted to the development of DEP-based LOC systems to promote their commercialization and wide utilization."
There should be more discussion on what the authors think needs to be done to promote DEP commercialization and wide utilization. What specific types of development is needed to move things forward?
We added discussion in conclusion as follows:
However, the wider employment of DEP would probably require higher level of system integration including sample treatment, post DEP processing as well as product analysis. Also the DEP itself should be designed more efficient way using 3D or semi 3D operations. That could be achieved either by designing the 3D electrode geometry or making the DEP area shallow suppressing the low efficiency space to promote the DEP chip commercialization and wide utilization.
Round 3
Reviewer 1 Report
The authors sufficiently respond to all questions raised by the reviewer. The manuscript is now acceptable for publication.